# Technology-assisted platform (TAP) for training and supervision of task-shared psychosocial interventions to ensure competency during scale-up in low-resource settings

Najia Atif[1] , Huma Nazir[1], Ahmed Waqas[2] , Anum Nisar[2], Hadia Maryam[1], Maria Kanwal[1], Maria Atiq[1], Mahjabeen Tariq[1], Ahmreen Koukab[1], Abid Malik[3] , Siham Sikander[2] and Atif Rahman[2]

[1]Human Development Research Foundation, Pakistan; [2]University of Liverpool, UK and [3]Health Services Academy, Pakistan

## Research Article

**Keywords:**
technology-assisted training; task-shifting; perinatal depression; non specialist providers; digital supervision

**Corresponding author:**
Atif Rahman;
Email: atif.rahman@liverpool.ac.uk

## Abstract

Common mental disorders are a major public health concern, particularly in low-resource settings where specialist services are limited. While task-shifting to non-specialist providers (NSPs) has improved access, maintaining their competency during scale-up remains a challenge. This study evaluated a technology-assisted platform (TAP) for training and supervision of NSPs delivering the WHO Thinking Healthy Programme (THP) for perinatal depression. The android-based hybrid platform integrates avatar-led instruction, digital modules, video demonstrations and structured supervision. Qualitative data were collected from three focus group discussions with peers ($n = 24$), one with trainers ($n = 4$) and four interviews with peers who left the programme. Data were analysed using the framework analysis approach. Peer competencies were assessed, in a simulated role play setting, using WHO's Ensuring Quality in Psychological Support (EQUIP) tools immediately post-training and at 6 and 12 months. The hybrid model, combining automated digital training with human facilitation, was well received. In-person trainers valued avatar-based instruction, video modelling and automated guidance. Participants reported high satisfaction with the digital learning experience, enhanced technological skills, knowledge retention and confidence. Structured supervision supported competency by standardising supervision agendas, case management and fostering ongoing learning. Competency scores demonstrated sustained improvement over 12 months. Technology-assisted platforms such as TAP represent a scalable and sustainable strategy for strengthening NSP training and supervision, helping to maintain and potentially enhance the competency of psychological intervention delivery in low-resource settings.

## Impact Statements

This study demonstrates the potential of technology-assisted training and supervision to transform task-shifting models for mental health care in low-resource settings. By integrating digital learning, avatar-led instruction, structured supervision and competency tracking into a single platform, the technology-assisted platform (TAP) enabled non-specialist peers to enhance their competency over 12 months. The model reduces reliance on scarce specialist trainers and supervisors, offering a potential scalable and cost-effective approach to sustaining competent delivery of psychological interventions and strengthening community-based mental health systems, particularly in resource-constrained settings.

## Introduction

Common mental disorders such as perinatal depression and anxiety remain a critical public health concern due to their high prevalence and the significant adverse effects these can have on both women and their children if not treated (Stein et al., 2014; Rogers et al., 2020). While task-shifting, the delegation of mental health care from specialists to non-specialist providers (NSPs), has played a key role in reducing the treatment gap (Van Ginneken et al., 2021), the pressing challenge now is *how to maintain NSPs' competence during scale-up.* Herschell et al. (2010) emphasise that training strategies alone, such as feedback and consultation, must be complemented by trained, skilled supervisors who themselves receive structured guidance. This combination is essential for enhancing therapist performance, increasing knowledge and supporting proficiency with complex procedures. However, ensuring NSPs' competency through adequate





training and skilled supervision can be resource-intensive and logistically challenging, particularly in low- and middle-income countries (LMICs), due to the limited availability of specialist trainers and the vast geographical areas in which NSPs operate. To address this challenge, we posed the question: *Can technological innovations be used to ensure NSPs' competency in the delivery of psychosocial interventions*? Emerging evidence suggests that digital tools, such as online platforms, apps and video conferencing, can be just as effective, if not more effective, than traditional training methods. A systematic review highlighted that technology-based training can improve NSP knowledge, skills and satisfaction, providing a scalable and sustainable solution for maintaining quality during scale-up, although challenges in scaling-up such tools remain (Singh and Reyes-Portillo, 2020).

We addressed this challenge by developing an integrated technology-assisted platform (TAP), which is a digital system using software, devices and communication tools such as videos to deliver training and supervision of NSPs in the World Health Organisation (WHO)-endorsed Thinking Healthy Programme (THP). The WHO provides detailed training manuals for THP in nine languages on its website. According to the World Health Organization (2015), potential trainers should ideally have received Thinking Healthy training and practiced it for 12 months in the community under supervision. The trainers are then evaluated for their competency, following which they can train NSPs in a cascaded model of training delivery (Murray et al., 2011). In a larger project of which this study is a part, the entire WHO THP manual was digitalised into an App following a detailed human centred design (HCD) approach, which is described elsewhere (Atif et al., 2022). Briefly, HCD places end users at the centre by grounding design in a deep understanding of their needs, goals and context. Guided by these principles, a multidisciplinary team of expert clinicians, end users (women with perinatal depression, husbands and community health workers) and a technology team was consulted throughout three iterative steps: (1) establishing context and user requirements, (2) usability testing of the prototype App with the design team and (3) usability testing with end users.

By embedding an interactive training and supervision modules within the App, we anticipated that this TAP could help NSPs (local lay women with lived experience of pre- and postnatal depressive symptoms called peers) to maintain and/or build their competencies over time.

### Description of TAP for training and supervision

#### THP-TAP interface and process

THP-TAP is a digital adaptation of the WHO-THP designed to support intervention delivery with integrated technology-assisted training and supervision modules for NSPs. Each module is accessible via clearly labelled buttons on the home screen. The intervention delivery module is available to NSPs, in-person trainers and supervisors, whereas the training and supervision modules are restricted to trainers and supervisors. TAP is server-connected and securely stores session data either in real time or when the device reconnects to Wi-Fi. It is designed to be user-friendly, requiring only basic literacy and no IT skills, and can be operated on a tablet device or smartphone.

#### TAP training

Upon logging into the training module, the interface displays the five training sessions (Figure 1). While the training is conducted by virtual "avatar specialist trainers" (Figure 2), the trainer plays a key

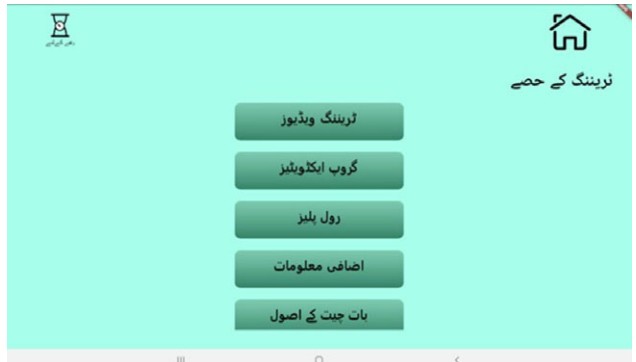

**Figure 1.** TAP interface displaying the training sessions. Here.

role in facilitating the process. Hence, both the in-person trainers and avatar trainers complement each other and make the process more real and interactive. The specialist avatars deliver brief lectures on psychoeducation, the core principles of the THP and cognitive behaviour therapy–based intervention delivery techniques. Furthermore, avatar trainers guide the in-person trainers through structured training instructions that appear at different time points, enabling them to facilitate group discussions and engage them in reflective dialogue. In-person trainers have access to brief videos which feature "mock" delivery sessions that allow them to model their delivery. Simulated role-plays are integrated throughout the module to reinforce learning, providing them with opportunities to practise their skills in a structured environment. The in-person trainer introduces a scenario, such as the first meeting with a participant, a challenging situation or an opportunity to demonstrate empathy, and asks NSPs to enact it in a role-play setting. Each role-play is followed by detailed, constructive feedback. When necessary, training videos are revisited to observe the application of skills and to allow further practice through additional role-play.

In summary, while avatar trainers play the key role in providing instructions and information, the in-person trainers facilitate the training sessions and support participant engagement by following detailed instructions provided within the training module. Human connection is therefore central in amplifying the effect, while fidelity to the training process is maintained through the App. During discussions, in-person trainers encourage NSPs to ask questions, seek clarification, request repetition of videos and receive feedback on their role-play performances. In addition, they, through the use of a predefined list of challenges, mitigation strategies and risk assessment questions embedded in the module, prepare NSPs to manage potential implementation challenges and identify signs of possible harm.

#### TAP training content

The digitalised training programme is designed to be delivered over a total duration of approximately 18 h. The content is based on the WHO-THP manualised training programme (World Health Organization, 2015) and is summarised in Table 1.

#### TAP supervision

TAP provides monthly supervision, which is conducted in a group format and facilitated by in-person trainers/supervisors, who are non-mental health specialists. The supervision module is accessible by selecting the "Supervision" button on the TAP home screen. Upon entering the module, the interface displays a password

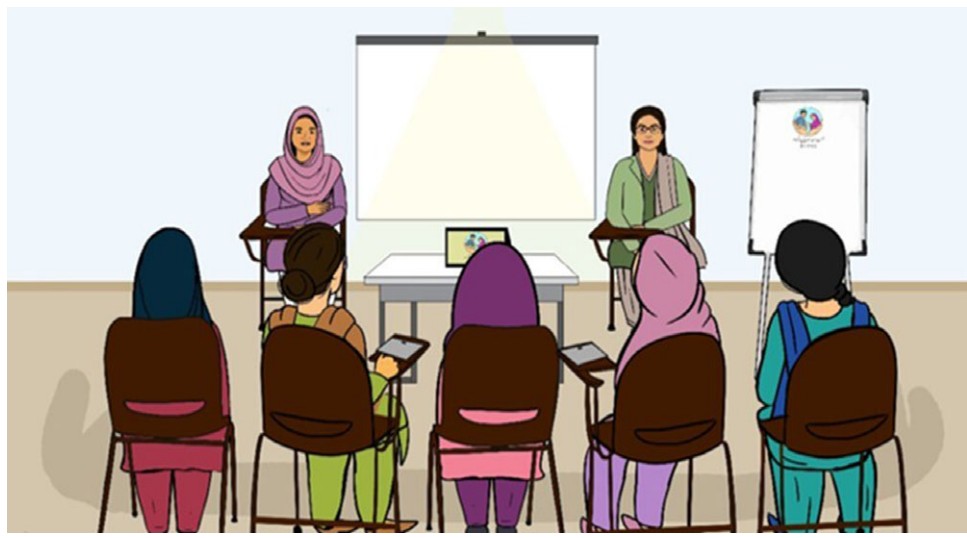

**Figure 2.** Avatars of trainers and peers. Here.

**Table 1.** Content and time allocation of the sessions

| Session | Sessions' content | Time allocated |
|---|---|---|
| Session 1: Familiarisation with tablet use and TAP features | Orientation includes hands-on practice with the tablet and the TAP features, focusing on key functionalities such as navigating forward, pause and back buttons, as well as accessing instructional slides. The session also introduces participants to the avatar specialist trainers and outlines the ground rules for the training process. | 1 h |
| Session 2: Training on the five core strategies of WHO-THP | Training on the five core strategies of the WHO-THP: building rapport and establishing an empathetic relationship, engaging family members, thought shifting to promote healthier thinking patterns, behavioural activation to encourage positive activities and problem-solving skills to address daily challenges. These strategies are reinforced through guided role-plays to enhance practical application and confidence in delivery. | 4 h |
| Session 3: Introduction to intervention content and delivery | This session introduces the core content of the intervention, focusing on the three key domains of maternal well-being: the mother's personal health, her relationships with significant others and bonding with her baby. Participants are also familiarised with the structured delivery mechanisms of the intervention sessions. The session includes guided role-play activities to help participants practice and internalise both the content and delivery approach. | 10 h |
| Sessions 4–5: Managing delivery challenges and ensuring recipient safety | This session focuses on equipping participants with the skills to address common challenges encountered during intervention delivery. It covers procedures for risk assessment, recognising and reporting adverse events and understanding the referral pathways for recipients requiring additional support. | 3 h |

protected login assigned to each in-person supervisor. By logging in to the supervision module, the supervisors can access personalised supervision data for each NSP they supervise, including the dates and durations of previous supervision sessions, their supervision attendance records. Additionally, information on the number of cases managed by each NSP, both completed and ongoing, along with session-level documentation of dates, duration and number of sessions delivered and their outcomes is available. App has inbuilt risk assessment flowcharts to help NSPs identify high-risk cases, which are automatically flagged within the system. TAP connection with the server helps systematically store all the data within the app, allowing for easy access to supervision logs and session records for continuous monitoring, review and addressing any identified risks.

By clicking the "Start Supervision" button, supervisors are guided through five distinct but overlapping supervision activities designed to promote consistency and quality in supervision. Similar to the training module, the supervision module has avatars of

specialist supervisors who guide in-person supervisors through each activity of the supervision process. These activities are summarised in Table 2.

Overall, the TAP supervision module provides a structured and interactive approach to potentially maintain NSP competencies and ensure consistent, high-quality programme implementation over time.

### Evaluation of training and supervision

This research explored the adoption and implementation of the TAP for training and supervision within a larger randomised controlled evaluation of the entire programme (Rahman et al., 2025), asking the key question: *Does TAP help maintain NSPs' competency in the delivery of the Thinking Healthy Programme?* Our study had two main objectives: 1) to investigate the experiences of NSPs and in-person trainers who were subjected to the TAP for

**Table 2.** Overview of supervision activities

| Steps | Supervision activities |
|---|---|
| Activity 1: Monitoring cases | • Review session logs to track the number and duration of sessions delivered.<br>• Monitor programme adherence and receive automated alerts for any reported adverse events.<br>• Address adverse events following established protocols. |
| Activity 2: Peer discussions | • Facilitate structured group discussions to share experiences and seek guidance.<br>• Support reflection on challenges and use the module's built-in list of challenges and mitigation strategies for troubleshooting. |
| Activity 3: Reinforcing core areas | • Revisit intervention contents using brief avatar-led lectures.<br>• Enable in-person supervisors and NSPs to review and reinforce core concepts to strengthen intervention delivery. |
| Activity 4: Role-playing scenarios | • Use video demonstrations to model effective session delivery.<br>• Engage in role-plays to practise delivery skills and enhance fidelity. |
| Activity 5: Supporting peer well-being | • Emphasise the importance of emotional resilience and encourage self-care practices.<br>• Foster sustained NSPs engagement and reduce burnout risk. |

training and supervision and 2) to evaluate the competency of NSPs in delivering the TAP 1 year after training.

## Methodology

### Study setting

This study was embedded within a cluster randomised controlled non-inferiority trial (cRCT) involving 980 perinatally depressed women, assessed at 3 and 6 months postnatal (Rahman et al., 2025). The trial compared the conventional face-to-face delivery of THP through community health workers call lady health workers or LHWs, trained and supervised by in-person specialist trainers in a cascaded training/supervision model, with THP delivered by peers with lived experience, trained and supervised using the TAP process, facilitated by the non-specialist in-person trainers and supervisors. The trial results are described in a separate publication (Rahman et al., 2025) and showed that THP delivered through TAP was not inferior, and more cost-effective, compared to the more labour-intensive THP delivered by LHWs closely supervised by specialists. This paper describes the qualitative process data about the experiences of the peers who received TAP training and supervision.

The study was conducted in rural areas of Kallar Syedan, Gujar Khan and Rawalpindi, Punjab, Pakistan. The literacy rate in the study settings is approximately 52%, with a lower proportion of women being literate compared to men (Pakistan Bureau of Statistics, 2021). Additionally, Pakistan faces a 38% gender gap in internet usage, with 45% of men and only 27% of women using the internet. A similar disparity exists in mobile phone ownership, where 81% of men own mobile phones compared to just 52% of women (GSMA, 2023). These disparities are even more pronounced in rural areas, where both internet usage and smartphone ownership are significantly lower than in urban populations.

### Participants' eligibility

A total of 50 peers with lived experience, aged between 19 and 45 years, with years of schooling ranging from 10 to 16 and no prior experience in delivering mental health care, were deployed in delivering the THP-TAP. Training and supervision were provided by four in-person peer trainers. They were aged between 25 and 44 years, had 14 to 16 years of education and between 1 and 2 years of experience working as peer supervisors in previous community-based mental health studies.

### Data collection and analysis for competency assessment

#### Data collection for competency assessment

To evaluate peer competency in delivering the THP-TAP, assessments were conducted at three time points: immediately after training, 6 months post-training and 12 months post-training. These assessments employed standardised simulated role plays developed for the WHO's Ensuring Quality in Psychological Support (EQUIP) platform, which is designed to enhance training and supervision for mental health and psychosocial support services. Assessors, who were trained in both the intervention and the assessment tools, used adapted role-play scripts suited for the cultural context of Pakistan. Each role play involved an actor portraying a depressed mother, assessors familiar with THP and the assessment forms and a peer delivering the intervention. Each session lasted approximately 30 min. These role plays were evaluated using two EQUIP tools: the THP competency form and the Enhancing Assessment of Common Therapeutic Factors (ENACT) (Pedersen et al., 2020). Both assessment forms, available through EQUIP, are widely used to determine whether NSPs meet competency benchmarks for safe and effective delivery, typically in standardised role-plays or observed sessions.

The ENACT form evaluates 15 domains of foundational helping skills, including non-verbal and verbal communication, rapport building, empathy, harm assessment, appropriate family involvement, collaborative goal setting, psychoeducation and feedback elicitation. Meanwhile, the THP competency form assesses 11 specific domains related to intervention delivery skills, such as mood and activity monitoring, psychoeducation on thoughts and behaviours, identifying unhelpful thoughts, generating alternative thoughts, reviewing homework and stress management. Each domain in both tools was rated on a four-point Likert scale, with level 1 indicating the presence of any unhelpful behaviours and level 4 reflecting mastery of both basic and advanced helping skills.

#### Quantitative analysis of competency assessments

Competency ratings from the ENACT and THP tools were assessed on four ordered levels (Levels 1–4) across domains. As these ratings are ordinal, analyses were descriptive. Frequencies and proportions of peers achieving each competency level (Levels 1–4) were calculated for each domain at each assessment time point (post-training, 6 months and 12 months) and are presented as percentages. To improve interpretability of overall competency trends over time, results were additionally summarised as the proportion of peers achieving Level 3 (proficient) and Level 4 (advanced mastery). These summaries were based on standardised role-play assessments and were not intended for hypothesis testing.

### Data collection and analysis for qualitative study

#### Data collection

Data were collected through focus group discussions (FGDs) and in-depth interviews (IDIs). A total of four FGDs were conducted with peers delivering the intervention ($n = 20$), one FGD with peer trainers ($n = 4$) and four IDIs with peers who left the programme. Purposive sampling was used to ensure diversity, including peers who either continued or discontinued their participation, those with varying educational levels (less than 12 years of schooling vs. 12 or more years) and those younger and older than 25. All trainers were included in the data collection process.

The data collection began in July 2023 and concluded in November 2023. Before participation, all individuals were briefed on the study's purpose, informed of their right to withdraw at any time and assured of confidentiality. Written informed consent was obtained from all participants. FGDs lasted between 60 and 90 min, while IDIs ranged from 45 to 60 min. These discussions and interviews took place at rural Primary Health Care Centres. Separate topic guides were developed in Urdu for peers and their trainers, then pilot tested and translated into English. These aimed to explore their experiences with the TAP - training and supervision. Two experienced female research fellows conducted data collection, with all sessions audio-recorded. Recordings were transcribed, and all transcripts were anonymised by removing participant identifiers. Identifiable raw data were securely stored in locked filing cabinets at the research site.

#### Data analysis

**Qualitative** data collection and analysis were conducted concurrently using the Framework Analysis approach, which offers a structured and systematic approach, particularly suited to health services research (Goldsmith, 2021). The analysis followed five steps: (1) familiarisation and identification of codes, (2) categorisation and development of thematic framework, (3) indexing, (4) charting of themes and sub-themes and (5) interpretation to uncover connections and relationship between themes. To ensure data accuracy and minimise the loss of finer details in translation, the analysis was performed on the original Urdu transcripts. To enhance the reliability of the findings, each transcript was independently analysed by two researchers. Weekly meetings were conducted with supervisors (NA, HN) to discuss findings, validate results and ensure consensus on the interpretations.

### Ethical considerations

Ethical approval was granted by the Research Ethics Committees of the Human Development Research Foundation, Pakistan, the University of Liverpool, UK and the National Bioethics Committee, Pakistan. All participants provided written informed consent before data collection.

This study adhered to the Consolidated Criteria for Reporting Qualitative Research (COREQ) guidelines.

### Results

### Competency assessment results

The first competency assessment was conducted the day after completion of training. All peers ($n = 50$) achieved at least Level 2 across all domains, indicating that none exhibited harmful behaviours immediately post-training, and could deliver the intervention under supervision. At this time point, no peers demonstrated Level 4 mastery, and only a small proportion (≤15%) achieved Level 3 across ENACT domains, indicating that competency was largely at a basic level immediately following training. As peers gained experience over time, competencies improved. At the 6-month assessment, approximately one-third of peers (30%) achieved Level 3 or higher across core ENACT helping skills, reflecting early progression to proficiency in some domains, while Level 4 ratings remained uncommon (≤15%). By 12 months post-training, competency increased substantially across domains, with the majority of peers (≥70%) achieving Level 3 or Level 4, and up to half demonstrating Level 4 mastery in selected ENACT domains, reflecting consolidation and advancement of foundational helping skills over time.

A similar pattern was observed for intervention-specific competencies assessed using the THP competency tool. Immediately post-training, no peers demonstrated Level 4 mastery and fewer than half (≤40%) achieved Level 3 across THP domains. At 6 months, around half to two-thirds of peers (~60%) achieved Level 3 or higher. By 12 months post-training, the large majority of peers (≥70%) demonstrated Level 3 or higher competency, with up to one-third reaching Level 4 mastery in domains such as stress management and communication skills. Figures 3 and 4 illustrate changes over time in non-specific and intervention-specific competencies, respectively. Detailed item-level scores are presented in Supplementary File 1.

### Qualitative results

Four FGDs were conducted, three with 20 peers and one with three trainers. Each FGD lasted between 90 and 120 min. In addition, five IDIs, lasting between 40 and 60 min, were conducted with peers who discontinued delivering the intervention. Data were collected until saturation was achieved. The median age of the peers was 32.5 years and they had at least 10 years of schooling. Peer trainers/supervisors had a median age of 34.5 years, with a minimum of 14 years of education and at least 1 year of work experience.

The qualitative data analysis resulted in the development of three main themes, each with two to three sub-themes (see Figure 5 below). These themes and sub-themes reflected the participants' and trainers' experiences of using the training and supervision module in which digital technology and human trainers worked together to deliver intervention training and supervision, demonstrating how the interplay between the two modalities enhanced and amplified learning.

Furthermore, data from peers who discontinued participation reflected experiences similar to those who remained in the study. Their discontinuation was primarily attributed to changes in their personal circumstances.

### Theme 1: Satisfaction with the TAP training

Ease of using new technology. Peers with no prior experience of using smartphones or tablet devices needed more time to get used to the devices, but the simple operational features did not pose major challenges in usability. Hands-on practice, including video demonstrations and practical sessions, significantly boosted their confidence. A trainer explained, *"Half of the initial training was spent familiarising peers with the device… By the end of the first day, around 80% felt confident using it."* (FGD with peer trainer 01, aged 29 years, 16 years of schooling).

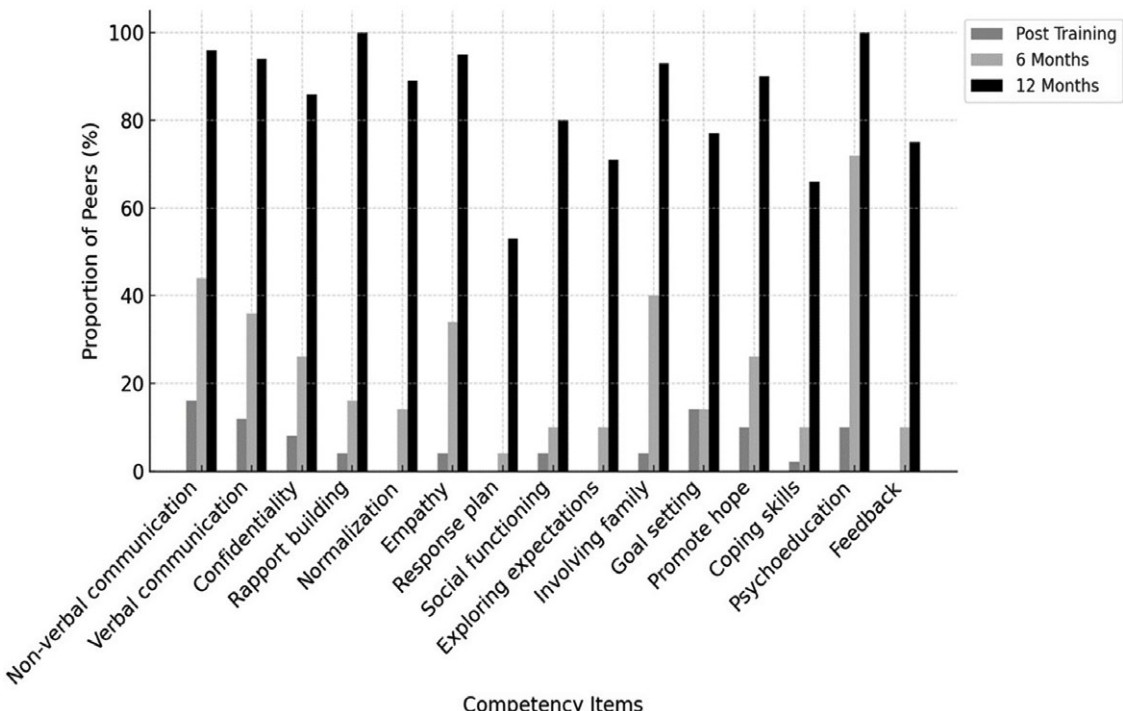

**Figure 3.** Competency assessment in non-specific therapy ingredients over time among the peers. Here.

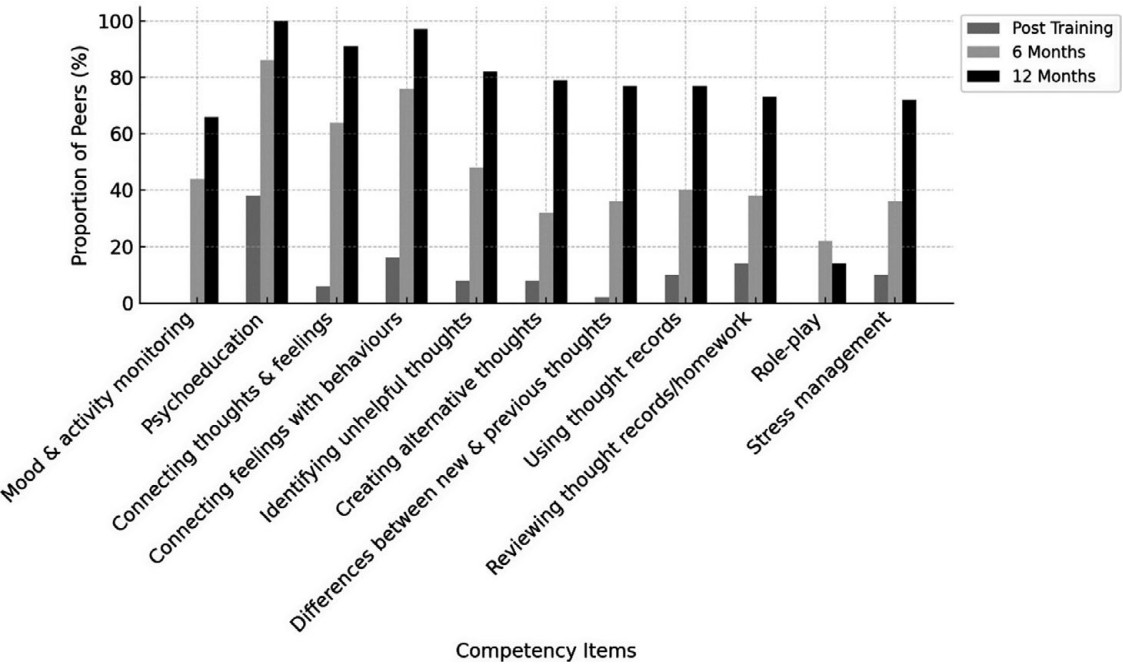

**Figure 4.** Competency assessment in specific therapy ingredients over time among the peers. Here.

Peers described the training environment as supportive, helping them move from apprehension about learning something new to confidence. *"What seemed impossible became manageable and even enjoyable thanks to the videos and (in-person) trainers support."* reported one peer (FGD with peer 07, aged 40 years, 10 years of schooling).

Some peers, already familiar with social media, felt motivated to explore digital options further. One said, *"Before the training, I used*

*my husband's phone for WhatsApp and Facebook. Afterward, I bought my own smartphone and started using other Apps."* (FGD with peer 02, aged 30 years, 10 years of schooling).

User-friendly hybrid learning model. The hybrid learning approach, featuring avatar-led videos and peer-facilitated role-playing, was well received by both trainers and peers. Trainers found the content easy to deliver, even without prior mental health work experience. A trainer shared, *"Everything was accessible in the*

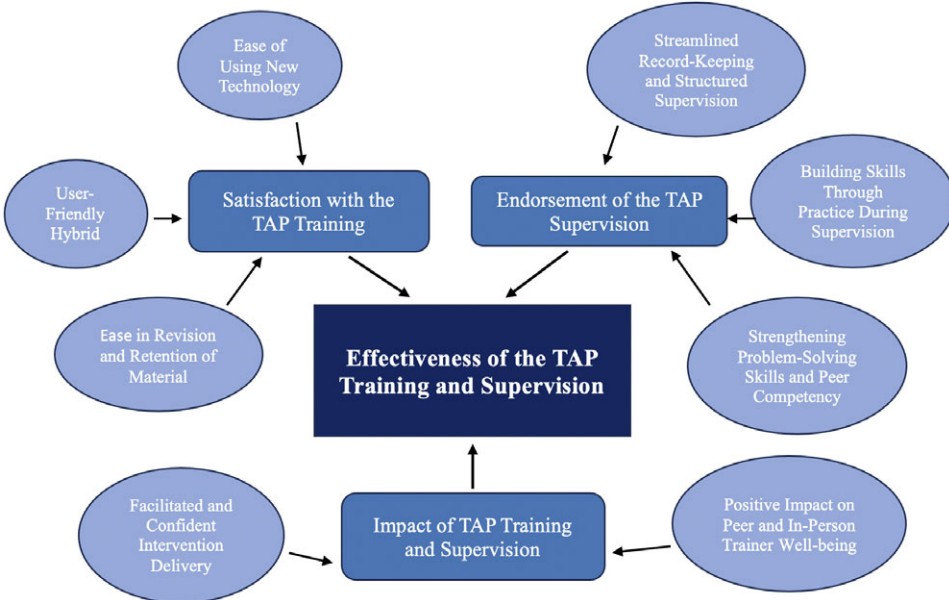

**Figure 5.** Thematic map showing themes and sub-themes from the qualitative data. Here.

*App. The avatar trainers explained the content simply and effectively."* (FGD with peer trainer 01, aged 29 years, 16 years of schooling).

Peers appreciated the practical guidance provided by the videos. A peer explained, *"Watching how the avatar peer interacted with the avatar mother gave us a clear idea of how to support depressed mothers."* (FGD with peer 04, aged 40 years, 10 years of schooling).

Some trainers, however, felt that more time for discussion would have enhanced understanding. *"A bit more time was needed for concepts that required deeper reflection,"* suggested one trainer (FGD with peer trainer 03, aged 41 years, 14 years of schooling).

Ease in revision and retention of material. Peers and trainers found the TAP training and supervision to be highly effective in reinforcing their learning. The engaging narrative-based video content was often compared to memorable TV dramas, making it easier to retain information. *"Technology-based training is like watching an impactful TV drama—it stays with you for years,"* said one peer (IDI with a discontinued peer aged 34 years, 14 years of schooling).

The ability to revisit training materials during training, as well as at home, further supported learning. *"If I forgot something, I'd just watch the videos again and it really helped,"* shared another peer (IDI with a discontinued peer aged 28 years, 10 years of schooling).

Trainers felt that the structured, avatar-based format with opportunities for modelling therapeutic skills, enhanced learning, especially for those new to mental health. *"First you watch and learn, then discuss and role-play. It was far more effective than traditional methods,"* reported one trainer (FGD with peer trainer 02, aged 30 years, 14 years of schooling).

### Theme 2: Endorsement of the TAP supervision

Streamlined record-keeping and structured supervision. Trainers and peers valued the TAP supervision for simplifying record-keeping and ensuring adherence to programme agendas. It enabled trainers to track attendance, monitor progress, flag high-risk cases and reduce errors. *"The portal-based system is far better than manual data entry, it reduces workload and chances of mistakes,"* noted one trainer (FGD with peer trainer 02, aged 30 years, 14 years of schooling).

Peers appreciated the App's password protection, which gave them confidence in maintaining confidentiality. A peer explained, *"If I had written notes, I'd worry someone might read them. With the password-protected tablet, I felt secure using it for supervision,"* (FGD with peer 15, aged 38 years, 10 years of schooling).

Pre-set supervision agendas also saved trainers time and allowed them to focus on critical issues, as shared by a trainer: *"There's no need to create agendas or take notes. That's a huge help when managing a high caseload,"* shared a trainer (FGD with peer trainer 02, aged 30 years, 14 years of schooling).

Building skills through practice during supervision. Supervision sessions helped peers overcome challenges and sharpen their skills through video reviews and role-plays. A peer recalled, *"We discussed mistakes openly. The supervisor had us rewatch training videos, do role-plays, and give feedback,"* (FGD with peer 04, aged 30 years, 10 years of schooling).

A key supervision component was risk assessment. The TAP for training and supervision guided peers with structured questions and flagged high-risk cases for review. *"In adverse events, the App prompted relevant questions, and we discussed flagged cases in supervision,"* explained a trainer (FGD with peer trainer 01, aged 29 years, 16 years of schooling).

Peers gained confidence applying these skills in real scenarios. A peer shared her experience of applying learning in a real situation. She said, *"When a mother quietly told me that she didn't have the will to live, I felt anxious but then I remembered the role-play. I followed the risk assessment steps and felt reassured when she said she had no plans to harm herself,"* (FGD with peer 15, aged 38 years, 10 years of schooling).

Strengthening problem-solving skills and peer competency. Supervision sessions enhanced peers' problem-solving abilities by encouraging them to discuss real-life challenges and explore solutions using the mitigation strategies suggested by the TAP and peer-trainer input. *"The solutions offered by the App, (in-person) trainers, and peers helped us address challenges effectively and build our problem-solving skills,"* said one peer (FGD with peer 07, aged 40 years, 10 years of schooling).

Trainers reinforced peer competency through continuous support, role-plays and content review. *"We stayed focused on challenges and used role-plays to maintain peer confidence and competency,"* noted a trainer (FGD with peer trainer 01, aged 29 years, 16 years of schooling).

Competency assessments using EQUIP and THP-specific tools validated improvements in competency. A peer proudly reported, *"Our competency improved from level 2 to level 4 because of regular supervision,"* (FGD with peer 06, aged 45 years, 14 years of schooling).

### Theme 3: Impact of TAP for training and supervision

Facilitated and confident intervention delivery. Training eased these concerns through observation of therapist avatars in a variety of real-life scenarios, helping peers model their own behaviour and responses and boosting their confidence. A peer reported, *"Watching the avatar therapist helped us become confident with the programme and have trust in our abilities,"* (FGD with peer 06, aged 35 years, 14 years of schooling).

Trainers also found the integrated TAP for delivering intervention instrumental in enabling lay peers to deliver the intervention effectively. A trainer said, *"The App gave me confidence that peers could deliver the content correctly,"* (FGD with peer trainer 01, aged 29 years, 16 years of schooling).

Prompts like the *"i"* button for quick information on navigating the App and the "beating heart" icon to remind peers to display empathy and use their helping skills, which enhanced delivery. *"These cues reminded peers to apply key skills and access information easily,"* shared a trainer (FGD with peer trainer 02, aged 30 years, 14 years of schooling).

### Positive impact on peer and trainer well-being.

Beyond skills-building, the training and supervision enhanced peers' and their trainers' personal well-being. They reported gaining insight into their own life-challenges and learning to manage them more effectively. This experience of personal growth is reflected in the quotes below: *"The training became a turning point for me, helping me overcome my own stress,"* (FGD with peer trainer 01, aged 29 years, 16 years of schooling). *"Before this programme, my life was full of crises, but I learnt to improve my life in many ways."* (FGD with peer 01, aged 29 years, 10 years of schooling).

The training also fostered empathy, deepening peers' understanding of pregnant women's struggles and strengthening their connection with participants. A peer compassionately reported, *"It gave me a better understanding of the issues faced by pregnant mothers,"* (FGD with peer 04, aged 30 years, 10 years of schooling).

All participants found role-plays improved confidence and communication, with some surprised by the positive feedback received during sessions. *"I was amazed when people praised my speaking. I wondered; did I really speak that well?"* said a peer (FGD with peer 02, aged 30 years, 10 years of schooling). Finally, the experience instilled a sense of purpose among peers. *"After four days of training, I realised it's not just about the stipend, it's about making a positive impact,"* affirmed a peer (FGD with peer 03, aged 27 years, 10 years of schooling).

## Discussion

Our study revealed that peers who received training and supervision through the TAP showed significant improvements in competencies twelve months post-training, assessed through simulated role-plays. This training model, integrated with technology-assisted delivery of the intervention and combining both technological tools and human facilitation, was well-received by both peers and their in-person trainers. In-person trainers appreciated the embedded "avatar" trainers for scaffolding intervention techniques, video-based vignettes for modelling delivery and automated cues for guiding role-plays. Peers reported enhanced technological skills, better content retention and the ability to easily access training videos for revision.

TAP supervision was instrumental in maintaining NSPs' competencies, ensuring adherence to supervision agendas, case management and flagging high risk cases. Both in-person trainers and peers acknowledged that the module contributed to smoother intervention delivery and improved psychosocial awareness, problem-solving skills, empathy and confidence. Additionally, supervision sessions helped address personal challenges and encouraged self-care, further boosting peers' satisfaction and engagement.

The findings of this study align with existing literature, which suggests that in-person training combined with technology can enhance participant satisfaction and skill retention. Similar to our results, other trials have shown that hybrid models that combine in-person facilitation with online components yield higher satisfaction compared to purely digital or online-only formats (Beidas et al., 2012; Rawson et al., 2013; Dimeff et al., 2015). Moreover, research indicates that expert-led in-person training tends to be more effective in improving competencies than self-guided study (Miller et al., 2004; Martino et al., 2011; Rawson et al., 2013).

Our study indicates that the avatar-based expert trainer was highly effective in teaching intervention techniques and helping skills through peer-to-peer interaction. Additionally, the importance of supervision in maintaining competencies is echoed in findings from previous studies. Enhanced supervision, when combined with role-plays and modelling techniques, has been shown to improve competency retention and intervention fidelity (Bearman et al., 2017). TAP for supervision does not require a specialist supervisor and can be facilitated in group settings by an experienced peer trainer. This is especially important for the scale-up of training and supervision, while at the same time ensuring competency is maintained.

Our study findings indicate that peers' competencies improved over time and reveal differential trajectories of skill development across competency domains (Figures 3 and 4). Non-specific competencies, as measured by ENACT, demonstrated a more pronounced increase between 6 and 12 months. This pattern likely reflects the core role of peers in providing empathy and supportive engagement, skills that are strengthened through cumulative practice and ongoing supervision and are broadly generalisable across helping contexts. These findings are consistent with existing evidence emphasising the importance of sustained support, practice and consultation in retaining and strengthening core competencies (Valenstein-Mah et al., 2020). In our study, the ability to revisit training materials and vignettes, alongside access to consultations with in-person trainers, appears to have contributed meaningfully to peers' continued skill development over time.

The TAP training was condensed into 18 h, spread over 3 days. In-person trainers and peers found this duration optimal, aligning with research suggesting that psychological training for perinatal depression can be effectively delivered in shorter periods, given resource constraints. Wang et al. (2022) emphasise that 3 to 5 days of digital training is considered an efficient time frame for interventions like perinatal depression treatment, particularly in low-resource settings where time and financial limitations often prevent longer training durations. Despite the reduced training time in our

study, participants expressed satisfaction with the delivery and reception of the training and supervision. Importantly, the embedded randomised trial showed that THP-TAP was not inferior to the standard WHO-THP and achieved greater reductions in depressive symptoms at 3 months postnatal (Rahman et al., 2025), further supporting the present study's finding that technology-assisted training and supervision supported NSPs' competency over the trial period.

Future studies should explore the optimal frequency and intensity of consultation and supervision sessions to maximise the retention of skills and fidelity to interventions. While our study did not measure the frequency of post-training consultations between peers and trainers, existing literature suggests that regular post-training consultations improve satisfaction and knowledge retention (Frank et al., 2020; Valenstein-Mah et al., 2020). Investigating the effects of different consultation models on long-term competency retention would be valuable.

Additionally, future research should assess how well the TAP training and supervision model can be scaled to different contexts, particularly in urban or more resource-intensive settings. Given that our study took place in rural Pakistan, replication in other cultural and geographical environments would help determine the generalisability of the model.

Finally, future work should assess fidelity through observing real-life sessions and could incorporate feedback on real-life session recordings as part of the supervision process, as this has been shown to further enhance fidelity to interventions. Testing how automated prompts for real-time feedback during intervention delivery influence training outcomes would also be a valuable addition.

### Strengths and limitations

One strength of this study is the dual assessment of competencies using two distinct tools recommended by the WHO EQUIP platform: the ENACT tool for Foundational Skills and the THP Competency Form for intervention-specific skills. This comprehensive evaluation provided a detailed understanding of both general and intervention-specific competencies. Additionally, qualitative feedback from both in-person trainers and peers, including those who discontinued delivering the intervention, strengthened the findings through triangulation. The use of framework analysis added further depth and rigour to the qualitative data analysis.

However, the study has several limitations. Potential assessor bias, such as leniency bias, when knowing that the peers are NSPs, during competency evaluations, may have affected the results, and the simulated role-play setting in which competency assessments were conducted may not fully capture the complexities of real-world intervention delivery. Data related to the training were collected approximately 1 year after its completion; therefore, participants' accounts reflect recalled experiences. Furthermore, the study's rural Pakistan setting limits the generalisability of the findings to other cultural and geographic contexts. Therefore, caution is advised when applying these results to different populations or environments.

### Conclusions

As digital tools become increasingly important in global mental health, models such as our TAP for training and supervision of non-specialist providers in LMIC settings offer an acceptable method for maintaining or even enhancing the competencies of these providers over time. For policymakers, the integrated training, supervision and delivery into a single digital platform provides a potentially scalable, resource-efficient strategy to strengthen community mental health services, particularly in low-resource settings where specialist availability is limited. Future research should focus on optimising supervision intensity, exploring scalability across diverse contexts and integrating real-time feedback mechanisms to further enhance training outcomes and intervention fidelity.

**Abbreviations**

| | |
|---|---|
| COREQ | Consolidated Criteria for Reporting Qualitative Research |
| cRCT | cluster randomised controlled trial |
| ENACT | Enhancing Assessment of Common Therapeutic Skills |
| EQUIP | Ensuring Quality in Psychological Support |
| FGD | Focus group discussion |
| HSD | Human-centred design |
| IDI | In-depth interview |
| LHW | Lady health worker |
| NSP | Non-specialist provider |
| TAP | Technology-assisted platform |
| THP | Thinking Healthy Programme |
| WHO | World Health Organisation |

**Open peer review.** To view the open peer review materials for this article, please visit http://doi.org/10.1017/gmh.2026.10202.

**Supplementary material.** The supplementary material for this article can be found at http://doi.org/10.1017/gmh.2026.10202.

**Data availability statement.** The data that support the findings of this study are not publicly available due to participant confidentiality restrictions but are available from the corresponding author upon reasonable request.

**Acknowledgements.** We would like to express our sincere gratitude to the peers who delivered the intervention to mothers experiencing perinatal depression, and to our team members, Naheeda Bibi, Mastoora Bibi and Faria Khan for their diligent transcription of the interview data. We would also acknowledge the use of ChatGPT (OpenAI, San Francisco, CA) solely for proofreading and language polishing. All content was reviewed and verified by the authors, who take full responsibility for the final manuscript; no patient or confidential data were entered into the platform.

**Author contribution.** Conception: A.R.; Design of the work: N.A., H.N., A.W., A.M., S.S., A.R.; Acquisition and analysis: M.K., M.A., M.T., A.K.; Supervision during acquisition and analysis: N.A., H.N., A.N.; Interpretation N.A., H.N., A.N., A.R.; Writing - original draft: N.A.; Writing - review and editing: H.N., A.W., A.M., S.S., A.R.; Funding acquisition: A.R. All authors have read and approved the final version of the manuscript.

**Financial support.** This research was funded by the National Institute for Health and Care Research (NIHR), UK (NIHR200817), using UK international development funding from the UK Government to support Global Health Research. The views expressed in this publication are those of the authors and not necessarily those of the NIHR or the UK Government.

**Competing interests.** The authors declare no conflict of interest. The funders had no role in the design of the study; in the collection, analysis or interpretation of data; in the writing of the manuscript or in the decision to publish the results.

**Ethics statement.** This study was conducted in accordance with established ethical guidelines and principles. Ethical approval was obtained from the Institutional Review Board of the Human Development Research Foundation, Pakistan. Informed consent was secured from all participants, including married adolescents, their families and trainers, ensuring they fully understood the study's purpose, procedures and their rights. Participation was entirely voluntary, and individuals were informed of their right to withdraw from the study at any time without any negative consequences.

Confidentiality and anonymity were rigorously maintained, with all personally identifiable information removed from transcripts and data securely

stored. Measures were taken throughout the research process to safeguard participants' comfort, dignity and emotional well-being, thereby minimising the risk of harm. This study demonstrates a strong commitment to ethical research practices, prioritising participants' rights and welfare while contributing valuable insights into mental health interventions for prenatal anxiety in low-resource settings.

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
