## [Reviewer Report]

Scalable methods for maintaining quality of care in task-shared psychological services are much needed and I appreciate the focus of this paper evaluating a platform to support training and supervision for specialists delivering a depression EBT. This technology moves the field forward in important ways and have made some suggestions to help improve this paper further:

Abstract:

- Could you briefly include some details to support the statement “structured supervision supported fidelity.” Is this referring to the competency scores?

Introduction:

- At the end of your first paragraph or beginning of your second, I would suggest citing several papers that document these issues with drop off in quality (which I agree with and is documented in the literature, I would just support that better here).

- It seems the development of this platform is described elsewhere (reference to Atif et al. 2022) but could you include a few sentences to summarize that process?

- Under eligibility, you discuss both THP-TAP and WHO-THP participants. I was finding this hard to follow and would suggest focusing here just on those discussed in this analysis, not the participants of the entire RCT.

- I would briefly describe any calculations done for the ENACT and the THP competency form - frequencies, proportions, etc.. under data analysis.

- When were focus groups conducted relative to training? How much later?

- Also how were these quantitative and qualitative data integrated? Did you look at the competency scores first? And then focus group content? Or vice versa?

Results

- Please include numeric data for your high level competency results rather than just “most” or “majority.” I had to turn to the figure to understand these data at all it felt.

- For themes, could you report demographic information instead of participant ID number - age, sex, etc...?

- I may have missed it but I did not see themes highlighted from those who discontinued. Can you address that more directly? It seems useful that you interviewed them.

Discussion

- What is the nature of the assessor bias you refer to? Did they know the participants? Please clarify that.

- In Figure 3, there is a notable jump in competency on many non-specific items from 6 to 12 months as compared to post-training to 6 months. It’s also interesting that it is not the case for specific items in Figure 4 (although average competence on these items seems lower than the non-specific items at 12 months). I was hoping for more commentary on these figures and how these data relate to your platform’s impact and what may be realistic to expect for NSPs in the discussion.

---

## [Reviewer Report]

The authors present an important and timely manuscript describing the development and use of the Technology-Assisted Platform (TAP) for training and supervision of nonspecialist providers (NSPs) delivering task-shared psychosocial interventions. This study addresses a critical challenge in global mental health—how to scale and sustain training and supervision for non-specialist providers. The manuscript contributes valuable insights; however, I think some revisions are needed to strengthen the conceptual clarity, expand the literature grounding, more fully describe the TAP model (particularly the integration of virtual and in-person components), and accurately interpret the competence-related findings. I believe addressing these issues will significantly enhance the manuscript’s rigor and potential impact.

Abstract and Impact Statement:

• The authors should clearly state that competency scores demonstrated sustained improvement but were not statistically tested. Without this clarification, readers may overinterpret these quantitative results.

• The manuscript frequently equates competency scores with “intervention quality.” Because assessments were based on role plays, not direct observation of clinical delivery, the abstract should specify that findings reflect provider competence in simulated practice, not fidelity or real-world quality. This distinction is essential and should be maintained consistently across the manuscript.

Introduction:

1. The introduction references a “quality gap” but does not provide evidence, definitions, or citations. The authors should expand this section with literature describing:

a. Whether a quality gap exists in global mental health (are there studies to document this quality gap in global contexts?)

b. How it has been conceptualized/defined

c. If global data are limited, evidence from high-income settings demonstrating fidelity and competence challenges could be used to illustrate the potential gap.

2. When citing Herschell et al. (2010), the authors should highlight a central conclusion: training alone is insufficient, and ongoing supervision or consultation is needed for skill transfer. This point meaningfully strengthens the rationale for TAP’s integrated training + supervision model.

3. Throughout the manuscript, TAP is described as a scalable or sustainable model; however, these findings represent early-stage implementation. Language should be revised to state that TAP is potentially scalable/sustainable, pending further evaluation.

4. These terms quality, competence, and fidelity appear to be used interchangeably but carry different meanings in implementation science. The authors should define each with citations.

5. The manuscript later mentions that THP was delivered by peers with lived experience, but this is central to the broader trial. A brief mention in the introduction would improve clarity and coherence.

Methods:

1. I feel the description of the TAP model and its components could be expanded. The current description does not sufficiently explain how virtual modules, in-person training, and on-the-ground NSP supervision fit together. I would have liked a more thorough description of which aspects of training were delivered through TAP, where in-person support was integrated, whether NSPs engaged in role plays with the live trainers or virtually, etc.

o The training content section should introduce and clearly describe the role of in-person trainers—how they supported or augmented virtual modules during initial training and throughout implementation.

o Some clarity on how role plays were conducted (in-person? with avatars? mixed modalities?).

2. The manuscript references training principles based on “NSPS-based learning” but does not define this term. I wasn’t exactly sure what that entailed or how it was defined.

3. Figures are included but not referenced in the Methods section. Each figure should be introduced and described when relevant.

4. The current description of TAP supervision focuses mostly on supervisors and how they can use the TAP platform. It is unclear whether NSPs use TAP during service delivery, whether they receive asynchronous feedback, if they submit role plays through TAP, or how/if TAP functions primarily as a supervisory tool.

5. Given their centrality to the qualitative analysis, the interview and focus group guides should be included as appendices.

Results

1. Please report whether statistical testing was performed to examine the change in competency scores. If no statistical tests were conducted, please include your rationale why and ensure to not overstate findings.

2. Please take care to not equate competence scores from role plays with real-world “intervention quality.” This should be emphasized throughout the results.

3. The qualitative findings are rich; however, without earlier clarity on what TAP does independently versus what trainers/peers provided, it was difficult for me to know what was specific to the technology platform v the assisted training. Some enhanced descriptions in the Methods will help readers interpret what results are specific to the TAP.

Discussion

1. The current discussion implies comparisons between TAP and self-guided models. Because the study did not include a comparison arm, the authors should revise language to ensure they only interpret what their own data can support, which is that TAP was acceptable and competence improved over time.

2. I would recommend avoid describing TAP as an “automated training model” as there is an in-person training and supervision component. The discussion should avoid implying full automation, which would overstate the current capabilities of the platform.

3. A full paragraph describing outcomes from the randomized trial in which TAP was embedded distracts from the manuscript’s main focus. I think it is appropriate to mention these findings in the introduction and briefly in the discussion but I think a more focused discussion on the findings of this manuscript is best.

4. The discussion should acknowledge that competence was measured in a simulated role-play context, not through observations of live delivery. The authors should continue to distinguish between competence, fidelity, and quality.

5. The final paragraph could be edited to emphasize some of the edits I have offered and focus the paper more directly on acceptability and (non-comparative) effectiveness of TAP.

Minor Comments

1. The authors should take care to define all acronyms. THP-TAP was a newly introduced acronym in the methods that made me wonder “is the TAP the intervention delivery tool or just a supervision tool?” An expanded description of the TAP itself and its training and supervision structures may address this.

2. The authors cite Singla et al. (2020) in the manuscript, but the reference list contains a Singh & Reyes-Portillo (2020) manuscript. Please double check your citations to ensure that you are citing all correctly in text and reference list.

---

## [Editor Report]

Thank you for submitting your manuscript for review. Although the reviewers acknowledge the relevance of the manuscript, they have identified notable flaws in the abstract, background, methodology, findings, and their interpretation. The reviewers have provided useful recommendations that could improve the manuscript. We invite you to carefully consider and address the reviewers’ comments and recommendations and submit a revised manuscript.

---

## [Reviewer Report]

I appreciate the continued efforts to revise the manuscript and further clarify the TAP model. The description of the training and supervision platform is improved, and the manuscript is moving in a positive direction. However, several conceptual and interpretive issues remain unresolved. These issues primarily concern the persistent conflation of quality, fidelity, and competence, and the interpretation of results beyond what the data can support. Addressing these points is essential to ensure conceptual rigor and accurate framing of the study’s contribution.

Conceptual Framing: Competence vs. Fidelity vs. Quality

The newly added paragraph acknowledging overlap between fidelity and competence must be revisited. These constructs should not be conflated. The results presented in this manuscript speak specifically to provider competence, as assessed through standardized role-play-based ratings. No direct data on fidelity of real-world intervention delivery or quality of care are presented.

As such, the introduction and discussion should consistently frame the study as examining development and sustainment of provider competence, rather than “quality” or “fidelity.” If the authors wish to situate competence as one component of broader quality frameworks, this can be stated cautiously, but I feel the primary framing should remain competence-focused.

The discussion of the Bond review is currently misplaced. This review addresses measurement of fidelity and competence, not conceptual arguments suggesting that divergence from fidelity means an intervention no longer functions as evidence-based. In light of the recommendation above to move away from fidelity framing, I encourage the authors to remove or substantially revise this section.

Finally, the continued use of “quality gap” language in the introduction remains problematic. This manuscript does not evaluate quality of care or fidelity of service delivery. Rather, it examines strategies to improve provider competence. The introduction should be reframed accordingly.

Methods

The revised methods would benefit from consistently specifying “Avatar“ or “In-Person” trainers or supervisors. This minor change to specify throughout could improve clarity.

While THP and TAP are defined separately, the combined THP–TAP model is not explicitly defined as a unified training and supervision system. A concise definition would help readers understand the integrated model being evaluated.

It remains unclear how TAP is used during actual intervention delivery. Because supervisors reportedly draw on TAP data to guide supervision, the manuscript should briefly describe how NSPs enter session data into TAP, whether TAP provides prompts or decision support during delivery, and how case tracking or risk identification occurs. Without this information, statements about TAP supporting case tracking or high-risk management are difficult to interpret.

Results

Theme 2: The label “quality control” again invokes a construct not measured in this study. I recommend renaming this theme using competence-focused language (e.g., “supporting and monitoring provider competence”).

Theme 3: The first quote in Theme 3 appears to describe TAP-supported intervention delivery, rather than training or supervision. Including this quote shifts the focus away from the stated aims of the paper. I recommend removing this quote to maintain thematic alignment with TAP training and supervision.

Discussion

The sentence: “The TAP supervision was instrumental in maintaining fidelity, ensuring adherence to supervision agendas, and facilitating case tracking and addressing high-risk situations.” is not supported by results presented in this manuscript. Fidelity was not measured, nor were case-tracking or high-risk management processes described in sufficient methodological detail. This statement should be revised to reflect findings specific to supervision processes and competence development only.

The statement: “Foundational, non-specific skills may consolidate earlier, whereas intervention-specific competencies require longer periods of supported practice to develop” overinterprets the findings. While results indicate that non-specific competence scores improved earlier than intervention-specific competencies, the term “consolidate” implies a developmental process that was not directly examined. A more neutral description of observed score trajectories would be more appropriate.

---

## [Editor Report]

Thank you for submitting your revised manuscript. The reviewer has identified several issues in the introduction, study aim, and methodology, along with helpful recommendations. A clearer statement of the study aim and keeping the narrative strictly confined to that scope would strengthen the manuscript. We encourage you to address these suggestions and submit a further revised version for evaluation.

---

## [Reviewer Report]

Thank you for your careful attention to my comments. I do believe the paper is much improved and do not have additional comments. Well done!

---

## [Editor Report]

Thank you for revising the manuscript and responding to the reviewers’ recommendations. The reviewers are satisfied with the revisions and have recommended that we accept the manuscript. We are happy to accept the manuscript in its present form and look forward to working with you through the publication process.